# OOCHIP: Compartmentalized Microfluidic Perfusion System with Porous Barriers for Enhanced Cell–Cell Crosstalk in Organ-on-a-Chip

**DOI:** 10.3390/mi11060565

**Published:** 2020-05-31

**Authors:** Qasem Ramadan, Sajay Bhuvanendran Nair Gourikutty, Qingxin Zhang

**Affiliations:** 1Agency for Science, Technology and Research, 2 Fusionopolis Way, #08-02, Innovis Tower, Singapore 138635, Singapore; nairs@ime.a-star.edu.sg (S.B.N.G.); qingxin@ime.a-star.edu.sg (Q.Z.); 2College of Science and General Studies, Alfaisal University, Riyadh 11533, Saudi Arabia

**Keywords:** organ-on-a-chip, microfluidics, cell co-culture, perfusion, silicon

## Abstract

Improved in vitro models of human organs for predicting drug efficacy, interactions, and disease modelling are crucially needed to minimize the use of animal models, which inevitably display significant differences from the human disease state and metabolism. Inside the body, cells are organized either in direct contact or in close proximity to other cell types in a tightly controlled architecture that regulates tissue function. To emulate this cellular interface in vitro, an advanced cell culture system is required. In this paper, we describe a set of compartmentalized silicon-based microfluidic chips that enable co-culturing several types of cells in close proximity with enhanced cell–cell interaction. In vivo-like fluid flow into and/or from each compartment, as well as between adjacent compartments, is maintained by micro-engineered porous barriers. This porous structure provides a tool for mimicking the paracrine exchange between cells in the human body. As a demonstrating example, the microfluidic system was tested by culturing human adipose tissue that is infiltrated with immune cells to study the role if the interplay between the two cells in the context of type 2 diabetes. However, the system provides a platform technology for mimicking the structure and function of single- and multi-organ models, which could significantly narrow the gap between in vivo and in vitro conditions.

## 1. Introduction

Pre-clinical drug screening aims to transform toxicity and efficacy testing from a system reliant on high-dose animal studies to one based primarily on human-relevant in vitro models. The phylogenetic distance between laboratory animals and humans, the discrepancy between current in vitro systems and the human body, and the limitations of in silico modelling, have generated the need for new solutions to the ever-increasing demand for safety screening of new substances. The inherent complexity of interconnected tissues in animal models makes it difficult to elucidate and track the physiological events that characterize the interaction between animal organs and exogenic factors. Therefore, improved in vitro models of human organs to predict drug efficacy, interactions, and disease modelling are crucially needed. 

While the simplicity of traditional in vitro models makes them robust and suitable for high throughput research, unfortunately, they provide only little biological relevance to the complex biological tissues of the human body. The validity of an in vitro model is dependent on how it reproduces the key physiological and biological characteristics of the in vivo system. Organs-on-a-chip (OOC) technology has a great potential to mimic many physiological properties of the in vivo organ systems. The ultimate target of the OOC technology is the development of effective and translatable in vitro models with emphasis on investigating physiological events that characterize the interaction between organs, the immune system and exogenic factors (i.e., pharmaceuticals, chemicals and nutraceutical stimuli) in health and disease states. Moreover, these systems are powerful tools to provide physiologically relevant in vitro disease models that faithfully reproduce the key physiological aspects of the complex human organs.

Microfabrication and microfluidics technologies provided the tools to create advanced cell culture systems which can be employed to mimic a specific tissue structure and create in vivo-like cellular microenvironments [1,2]. They allow control of micro-environmental parameters, such as cell–cell and cell-matrix interactions. Perfusion-based media supply allows for the delivery and removal of soluble molecules into the cell microenvironment and controls the application of shear stresses by the fluid flow. These tools are, therefore, well suited to the study of biological interactions down to the cell and molecular levels and have tremendous potential to be applied to studying human physiology and pathology. 

The microfluidic structure in OOC devices can be designed to provide tunable dynamic fluid flows as well as delivering nutrients, chemicals, and stimuli to cells and the removal of soluble molecules with high spatiotemporal resolution in a controlled manner. This controlled dynamic flow would also provide mild shear stresses on cell membranes within the physiologically accepted ranges. 

Inside the human body, cells are organized in a unique heterogeneous architecture either in direct contact or in close proximity to other cell types in a tightly controlled architecture that regulates tissue function. Cells also interact with various environmental factors, such as the extracellular matrix (ECM), soluble molecules, including growth factors, hormones and cytokines. These chemical and physical factors are responsible for tissue organization during development and help to maintain cell functions. Homotypic and heterotypic cellular interactions are essential for tissue development, repair, and homeostasis [3]. OOC comprises a co-culture of more than two cell types to mimic the structure and function of a specific organ or multi organs. The simplest co-culture system permits direct physical contact between heterotypic cells [4,5] by combining different cell suspensions at the desired co-culture ratio to study the paracrine signaling and response to soluble signaling factors [6,7]. However, it is difficult to determine the relative contributions of each cell type to any observed effects of co-culture; particularly, when the impact of paracrine signaling are of interest [3]. To allow better differentiation of the heterotypic cellular signaling, a physical barrier, such as a membrane, between cell types is utilized that permits the exchange of soluble factors, while still preventing cell–cell direct contact. 

Modeling of human physiology and diseases with interconnected microphysiological systems (MPSs) or body-on-a-chip models is emerging as a prominent frontier in medicine. The last few years have witnessed several attempts to demonstrate complex MPSs with higher-order tissue functions [8,9,10,11,12,13,14,15,16,17,18]. Inter-organ communication is mainly studied through a systemic common medium that interconnects different organ modules to mimic the circulating blood, which can mediate organ-organ crosstalk. Despite the tremendous achieved progress, the existing technology gap between the current MPS technology and in vivo conditions is still wide. Particularly, the employed methods for emulating cell–cell cross talk are not capable of mimicking those in vivo. The vast majority of OOC and MPS devices are fabricated using soft lithography techniques out of polydimethylsiloxane (PDMS) which absorbs small organic compounds, including the precious analytes. In addition, using these techniques, one may not be able to achieve precise positioning of different types of cells in close proximity to a reliable and repeatable manner for efficient and physiologically relevant paracrine signaling. 

Hung et al. (2004) [19] described cell perfusion through porous barriers in a microfluidic chip; however, the cell culture array chip hosts only a monoculture of HeLa cells. Toh et al. [20] described a microfluidic-based 3D culture system that allows cell–cell and cell-matrix interactions by employing an array of micropillars that separate three parallel microfluidic channels and enable continuous perfusion. A similar system was recently used to study cancer cell migration and invasion across the basement membrane [21]. These devices were fabricated from PDMS with parallel and straight microfluidic channels with simple geometry and very large perfusion channels (i.e., the spacing between pillars) of 20-µm width. Constructing complex tissue architecture in vitro was pioneered by the work of Bhatia et al. [22,23,24,25], which enabled versatile microscale organization of multiple cell types, however, with regular patterns.

In this paper, we describe a micro-engineered cell co-culture system that enables the recapitulating of the structure of an individual human organ or a network of organs so that metabolites and paracrine signals can be transported and exchanged between various tissues or organ models. The modular perfusion system is a compartmentalized fluidic system that relies on thin porous walls or barriers which separate different fluidic compartments. Each microfluidic compartment physically retains a single cell type while permitting chemical signaling with the neighboring compartments through diffusion and pressure-driven flow. This configuration emulates the highly vascularized tissue in the human body where tissues interact directly with cells in the bloodstream (e.g., immune cells) and those in other tissue types. The compartmentalized fluidic structure enables the co-culturing of various types of cells in a planar organization. The circulation of cell-culture media across the multicompartment system carries the soluble factors released by the cells which could be collected for on-chip/off-chip analysis (Figure 1a). The system is designed with micro-engineered features that allow to control the fluidic characteristics through a compartmentalized bioreactor which is able to host various cellular modules to recapitulate various physiological and metabolic characteristics in the human body. These include:(1)Maintaining a reasonable and physiologically relevant tissue/tissue and tissue/liquid ratio between the different organs. This is achievable due to the miniaturized nature of the microfluidic structure and the continuous perfusion of cell nutrients and stimuli without compromising the reagent cost. For example, when cells are cultured in a conventional culture flask with a surface area of 75 cm^2^, 10 mL of culture media is added onto the cell culture which results in a media/cell ratio (volume/volume) of 67 times. On the other hand, when the same cells are cultured in a microfluidic chip with a surface area of 1–2 cm^2^, the volume of media covering the cells is within the range of 20–30 μL which results in a ratio of media/cell of 6 only.(2)Controlling spatiotemporal fluidic flow into and from the cell culture compartments to emulate the local blood flow and mass transport of nutrients into and out of the tissue.(3)Maintaining physiologically relevant shear stresses onto the surface of tissue.(4)Allowing micro-scale spatial heterogeneity of cells and tissue.(5)Facilitating efficient cell–cell and tissue-tissue interaction between heterogeneous cell cultures.(6)Enabling the spatial organization of different types of cells/tissue in 2D architectures with a physiologically relevant order, which may allow to track the key physiological events in disease progression and treatment.(7)Allowing the recruitment of circulating immune cells with a high spatiotemporal resolution of cell–cell interaction.

After fabrication and characterization, the system was tested with a parenchymal-immune cell co-culture model, namely adipocytes-immune cells (Figure 1b). 

Constructing a physiologically relevant biological model within a microfluidic system enables a quantitative analysis of the body interaction with exogenous substances and an understanding of the physiological events and pathways in disease pathogenesis. 

## 2. Materials and Methods 

### 2.1. Design and Simulation 

The goal of microfluidic-based perfusion technology is to shrink the gap between the in vivo and in vitro conditions. Owing to their small size and the ability to precisely control the fluid flow within these devices, it becomes possible to mimic specific key features of the tissue structure and cellular interface by positioning two or more cell types in close proximity to one another and the desired order or orientation. Combined with the fluidic perfusion, which maintains the mechanical shear stress on the cell surface, mass transfer of a cell’s nutrients and paracrine and endocrine signaling, these would provide the necessary tools to mimic in vitro the key functions of the tissue/organ of interest. Therefore, the microfluidic features, including channel/compartment size, the ratio between the various compartments within the device, the ratio between the volume occupied by cells and the volume occupied by the fluid, the distance between the different types of cells, and the flow profile within the cell culture compartments, are very important parameters that need to be carefully optimized in any OOC device. Figure 2 shows a 3D schematic drawing, optical images and SEM images of key features of the fabricated devices. Four microfluidic systems are described in this paper; however, the chip can be custom designed to host the desired tissue/organ model with a cellular orientation/distribution that mimics the in vivo counterpart. The planar geometry of these chips takes into consideration the order/positioning of any cell type with respect to the other neighboring cell type and the direction of the flow/perfusion where the direction of the perfusion flow defines the interaction between the cells/tissue within the various compartment. Briefly, the following chip designs are: Chip 1 (C1): Three circular concentered compartments separated by porous barriers (Figure 2a). This chip was used in our previous publication with a focus on adipose tissue in vitro models [19].Chip 2 (C2): Two circular concentered compartments interface with a downstream semi-circular compartment (Figure 2b).Chip 3 (C3): Four U-shaped concentered compartments with the inner compartment connected through a channel to another circular compartment (Figure 2c).Chip 4 (C4): Two interdigitated compartments interfaced with a small circular compartment located inside the upstream compartment (Figure 2d).

Different types of cells can be cultured in a two-dimensional (2D) organization within these planar compartmentalized structures where the multicompartments are physically separated by semi-porous barriers. In such cellular organization, each cell type is nested in its corresponding permeable compartment. Therefore, cells of different types are fluidically and chemically connected. A large number of the capillary surrounding each compartment facilitate efficient mass transfer into the constructed tissue, which mimic the physiological flow in the human body (e.g., blood flow in capillaries), and consequently prolong the tissue (i.e., organ model) viability for multiparametric analysis. The porous barriers are characterized by an array of small pores which are located in the upper side of the walls. The size of these pores is customized to retain the cells in their corresponding compartment and enable the exchange of the circulating soluble factor within the extracellular matrix (ECM) between the different compartments with fluidically controlled mass transfer between the various compartments. Each pore within the wall creates a short microchannel, called a perfusion microchannel, which connects the two compartments through perfusion. The ratio between the microchannel depth and the compartment depth is 5/200. The rate and direction of the intercompartment flow can be controlled using an external pump; hence, the heterotypic cell chemical interaction can be emulated by microflow through the porous barriers. In this setting, the microchannel array at both sides of any given compartments provides a chemical gradient of the solution with the adjacent compartments. The composition and concentration of chemicals within the side compartment can be selectively adjusted by controlling the flow rate through the upstream compartment. 

### 2.2. Chips Fabrication 

The chips were fabricated in silicon substrate using standard lithography and etching techniques (Figure 3a). For simplification, a cross-sectional drawing of a small part of the chip is shown. A bilayer of silicon oxide and silicon nitride was deposited on Si substrate with a thickness of 300 Å and 1500 Å (Figure 3ai), respectively. Using the deep reactive ion etching (DRIE) technique, shallow trenches with width, length and depth of 5 µm, 50 µm and 5 µm, respectively, were created to form the microchannel array of the porous barriers between the fluidic compartments (Figure 3aii). Afterwards, wide trenches, with a depth of 200 µm, were also etched using DRIE to form the fluidic compartments as well as the fluidic inlets/outlets (Figure 3aiii). The fluidic inlets and outlets were opened by further backside wet etching such that the fluidic port would be accessed through the backside of the chip (Figure 3aiv). Following this, the patterned Si wafer was bonded to a glass substrate using anodic bonding to form a fully sealed compartmentalized microfluidic structure with porous barriers. Finally, the silicon wafer was diced to individual chips with dimensions of 40 × 15 mm. The distance between the microchannel array and the surface of the cell culture compartment is 195 µm; therefore, media flow though these channels ensures mild shear stress onto the cell membranes.

### 2.3. Perfusion Setup 

The chips were accommodated in a custom-made fluidic connector, which was fabricated in poly (methyl methacrylate) (PMMA) to enable long-term perfusion during the cell culture (Figure 4). The serializable fluidic connector has L-shaped fluidic channels with a diameter of 1 mm to which the fluidic inlets/outlets of the chip were aligned (Figure 4). The chip is inserted in a shallow trench (1 mm) with the holes (fluidic ports) facing the holes in the trench. To prevent any liquid leakage, a PDMS gasket was placed between the chip and the connector surface and the chip was pressed against the connector using a PMMA frame and screwed to ensure a good connection. The fluidic connector also features vertical holders to hold the media reservoir. The fluidic inlets were connected to 15 mL tubes containing the cell culture media through Polyether ether ketone (PEEK) fittings (1/32 in., M-645X, ThermoFisher) and tubing and the fluidic outlets were connected to 10 mL syringes which were mounted on a set of high-precision syringe pumps (neMESYS, Cetoni, Germany) and controlled by QmixELEMENTS software (Cetoni, Germany). Finally, the perfusion setup was completed by placing the chip (within the holder) and the media reservoir inside a CO_2_ incubator and the syringe pumps were placed outside the incubator (Figure 4). The syringe pumps were run in withdrawing mode such that cell culture media perfused through the chip from the media reservoir which was maintained within the same culture conditions inside the CO_2_ incubator while the pumps were kept outside the CO_2_ incubator. Cellular supernatants were collected from the syringes whenever needed.

### 2.4. Finite Element Analysis

Finite element analysis (FEA) was carried out using the COMSOL Multiphysics software. A three-dimensional model with geometry that imitated the layout of the chips was created and the fluidic boundary conditions, materials and physics were applied. A laminar flow interface was used to compute the velocity of the fluid by solving the Navier–Stokes equations. Various conditions corresponding to the input/output ports were simulated to investigate the flow dynamics through the compartments by selectively applying the fluid flow through individual inlets. The velocity field and shear stress profiles were calculated at a fluid (water) flow rate of 8 nL/s. 

### 2.5. Characterization of Inter-Compartment Permeability 

To examine the mass transfer between the compartments, Fluorescein isothiocyanate (FITC)-dextran with a molecular weight of 4 kD was used as a tracer. FITC-dextran was diluted at a concentration of 5 µg/mL and injected into the corresponding compartment. The fluidic inlets and outlets (I/O) were opened/closed to allow the tracer molecule to be transported from a particular compartment to the adjacent one only through the porous barriers, either by diffusion or pressure-driven (convective) flow. The fluorescence intensities (FI) of the FITC-dextran within the compartments were monitored under a fluorescent microscope (BX63, Olympus, Tokyo, Japan). The fluorescence intensity indicates the level of porous barrier permeability and the intercompartmental mass transfer. The mass transfer of FITC-dextran tracer was monitored over 2 h under diffusion and convective flow with a flow rate of 20 µL/min in two parallel experiments for each chip. 

### 2.6. Cell Culture 

Cell culture/co-culture was demonstrated on the chip using human pre-adipocytes (#802s-05a, Cell Applications, Inc., San Diego, CA, USA) and U937 mononuclear cells (American Type Culture Collection, USA) as cell models. These cells were previously employed to construct an in vitro model of the inflamed human adipose in our recent publications [26,27] to study the interplay between immune cells and adipocytes in human obesity and insulin resistance. Obese tissues are heavily infiltrated by inflammatory immune cells (e.g., monocytes, macrophages, Th1 cells) which interact with adipocytes to trigger chronic inflammation, ultimately leading to blockage of insulin action on adipocytes and insulin resistance [28,29]. The interaction between the human adipocytes and immune cells and tissue-resident macrophages represents an example of how the interplay between cells contributes to pathogenesis (in this case, type 2 diabetes). 

Prior to cell inoculation, the fluidic compartments of the chips were filled with 70% ethanol for at least 4 h for sterilization. Then the chips were washed with deionized (DI) water and 100 µL of Poly-L-lysine solution (0.1 mg/mL, in H_2_O) was loaded into each chip and incubated overnight at 37 °C. The chips were then washed with sterilized DI water and phosphate-buffered saline (PBS) and primed with a preadipocyte growth medium (Cell Applications). 50 µL of Human Preadipocytes (HpA) suspension (cell density > 1.5 × 10^6^ cells/mL, cell viability > 85%) was inoculated into one compartment. Cells were maintained under static culture conditions (without perfusion) for 2 h to allow cell attachment in 5% CO_2_ humidified atmosphere. Then perfusion was started at a flow rate of 8 nL/s. The preadipocytes grew to confluence after ~2–3 days. To induce cell differentiation, the preadipocyte growth medium was replaced by adipocyte differentiation medium (#811D-250, Cell Applications). After 12–14 days of differentiation, an adipocyte maintenance medium (#811M-250) was applied for at least two days before cell characterization or co-culture with immune cells. Cell morphological images were acquired with an upright microscope (Olympus, BX3-CBH, Japan). After adipocyte differentiation, U937 cells were collected and suspended in an adipocyte maintenance medium. 50 µL of U937 cell suspension was inoculated into the adjacent compartment. The adipocyte maintenance medium was perfused at a rate of 8 nL/s through the chip for 2–3 days. Cell viability and morphological change were monitored by a microscope. A mixture of calcein-AM (2 μM) and ethidium homodimer-1 (4 μM) diluted in Dulbecco-PBS (D-PBS) was used to test the cell viability under microscope. Cell viability was quantified by the percentage of calcein-AM-labelled cells.

### 2.7. Lipid Droplets Staining and Quantification 

To quantify the amount of lipids within the adipocytes as an indicator of cell differentiation, cells were stained with Oil Red O (Sigma-Aldrich, St. Louis, MO, USA), which is used to stain lipids. Cells were first fixed with 4% paraformaldehyde (PFA) and treated with Oil Red O solution for 30 min at room temperature and then washed with DI water. Bright-field images were taken using a microscope. The total surface covered with lipids was estimated by taking several images of the adipocyte culture compartment in monoculture and analyzed using ImageJ software. 

### 2.8. Immune-Metabolic Profile/Cytokine Secretion-Glucose Uptake 

Adipocyte/U937 cells were cultured on-chip, as explained above. The inflammation state was induced by treating the cells with 100 ng/mL of lysophosphatidic acid (LPA). Supernatants were collected from the sampling outlet. Interleukin-6 (IL-6) concentration in the supernatants was measured using ELISA, as explained elsewhere [27]. For control, experiments were also conducted on untreated samples and adipocytes monoculture. Glucose uptake by the differentiated adipocytes was measured using the glucose uptake assay kit, which uses 2-deoxy-2-[(7-nitro- 2,1,3-benzoxadiazol-4-yl)amino]-D-glucose (2-NBDG) as a fluorescence-labelled deoxyglucose analog probe (Cayman Chemicals, Ann Arbor, MI, USA) [26].

## 3. Results and Discussion 

### 3.1. Finite Element Analysis

The perfusion profile in the compartmentalized chip was simulated with FEA at selected flow configurations by injecting water into selected fluidic inlets and opening selected outlets which correspond to perfusion during the cell incubation time (Figure 4a). Figure 4b shows the FEA meshed models with fine mesh implemented at the porous barriers. The velocity field profile and streamlines (Figure 2c,d) show high velocity near inlets/outlets and within the narrow compartments, while the flow velocity profiles tend to be uniformly low within the wide compartments. In general, the velocity observed was at its highest within the sampling compartment of Chip 4 and at its lowest towards the end of the interdigitated fingers of the same chip, while the velocity profile is more uniform across the culture compartments thanks to the perfusion channel arrays. The corresponding shear stresses on the cell membranes were calculated along a central line across the cell culture compartments (Figure 4e). The resultant shear stresses in all cell culture compartments were found to be within the range of in vivo physiological interstitial shear stress (≤0.1 dynes per cm^2^) [30]. It should be noted that shear stress can be adjusted by maneuvering the flow rates within the various compartments. 

### 3.2. Microfluidic Testing: Inter-Compartment Permeability

The permeability between the adjacent compartments was investigated using FITC-dextran 4k Da tracer. The tracer was injected into one compartment (upstream) at a concentration of 5 mg/mL until filled with the tracer (observed under the microscope). Then the tracer syringe was replaced by a clear PBS solution and the perfusion was set at a flow rate of 50 µL/min. The fluorescence intensity was recorded at different time intervals across each chip compartment. Figure 5 shows selected fluorescent images that indicate the tracer transport through the compartments over a period of 90 min with a time interval of 10 min. Figure 5b shows the permeability profile between two adjacent compartments as indicated by the fluorescence intensities. The recorded intensities and the zone of interest are marked in the inset within the corresponding chip. The FI was measured when the compartments were filled with a clear PBS solution (no FITC-dextran) and subtracted from the total apparent FI to obtain the actual FI due to the dextran permeability. 

### 3.3. System Testing with Parenchymal-Immune Cell Co-Culture

The OOC chips were tested with an in vitro model that was established and reported in our previous publication [26], where the experimental procedures are also described. In brief, the biological model comprises a co-culture of human adipocytes (parenchymal) and immune cells which are positioned in two adjutant porous compartments in dynamic culture conditions. Four co-culture experiments were conducted in parallel using the four different chips. Human pre-adipocytes (HPADs) were initially seeded in the designated compartments (Figure 6a,b) and the cell growth and morphology were regularly monitored under the microscope. After HPADs reached confluence ~3–5 days after seeding (Figure 6c), the cells were perfused with a differentiation medium, and lipid droplets were observed after ~2 days (Figure 6d). The differentiated adipocytes are characterized by the presence of different sizes of lipid droplets. Droplet size (hypertrophy) and number (hyperplasia) gradually increased over the 14 days of the differentiation period (Figure 6e–g). To measure the adipocytes hyperplasia, the intracellular content of the cells was stained with fluorescence to enable observing the cell boundary. The lipid droplet number continuously increased during the differentiation period (Figure 7a). Hyperplasia was clearly observed during the first week of differentiation of all chips with a significant number of droplets found in Chips 1 and 2 (Figure 6). However, during the next two weeks, the individual cell size tended to be increasingly enlarged due to the increase of the lipid droplet size (i.e., the hypertrophy is dominated over the hyperplasia). Figure 7b shows the highlighted lipid droplets within the adipocytes, which were stained with Oil Red O stain, and Figure 7c shows the total lipid coverage within the adipocyte compartment which is correlated to the total lipid contents of the adipocytes after 5 and 24 days of adipocyte monoculture. These data confirm the differentiation of preadipocytes to adipocytes (adipogenesis) on-chip. Adipocyte growth and differentiation in co-culture mode were similar to those in monoculture mode. No difference was optically observed. 

For immune-metabolic analysis, adipocytes and immune cells were cultured in their designated compartment (Figure 6a) and the immune-metabolic analysis of the co-culture in Chip 2 was conducted after two weeks of co-culture. By comparing the fluidic and cell co-culture characteristics within the four chips, despite the observed similarity, Chip 1 and Chip 2 showed more uniform fluid flow (Figure 4c,e) and were less prone to bubble generation that could interrupt the cell culture experiments. Due to some chamber narrowing in Chip 2 and Chip (4), the fluid velocity dropped at certain area, such as within the outer compartment of Chip 3 and at the circular ends of the interdigitated fingers of Chip 4. Therefore, we only present the immune-metabolic results of Chip 2. 

The main objective of the current paper is to test various planar arrangements of porous compartments for cell co-culture and to demonstrate a set of designs that can be adapted to selectively position different types of cells in close proximity with efficient intercompartmental crosstalk for organ/disease modelling. Therefore, we only demonstrate the basic functions, i.e., cell co-culture, differentiation and the basic immune-metabolic profile of the co-culture. In-depth characterization and screening of the biological model was partially reported in our previous publication and is also currently being conducted. 

The key feature of our microfluidic system is its ability to host a co-culture of two or more cell types in close proximity, which are physically separated but continuously exchanging bio/chemical signals similar to in vivo environment. 

The tested in vitro model mimics the immune cells infiltrate into the human adipose tissue, which is associated with obesity [31] and insulin resistance [28]. It was reported that human adipose tissues express a number of inflammation-related genes [32,33] with obese subjects displaying a higher expression of these genes as compared to non-obese subjects [34]. These findings support the growing importance of immune-metabolic interplay in human obesity that link to insulin resistance and type 2 diabetes mellitus (T2DM).

Adipocyte/U937 were cultured on-chip as explained above and the inflammation state was induced by treating the cells with 100 ng/mL of LPA. Then, the glucose uptake by the adipocytes and the secreted TNFα and IL-6 (in supernatants) were quantified from the same cell co-culture set. Figure 7d shows the combined immune-metabolic profile of the co-culture. A good correlation between the cytokine profile and the glucose uptake was observed. In adipocyte monoculture, a significant glucose uptake was observed with no cytokines upregulation being detected. However, in the co-culture set, a slight decrease was observed in glucose uptake with a notable increase in cytokine secretion, particularly IL-6. On the other hand, when the inflammation state was induced by treatment with LPA, a significant increase of cytokine secretion and decrease of glucose uptake was observed, which is in agreement with our previous observation [26,27]. 

The design of the chips enables long-term dynamic cell co-culture that enhanced the cell–cell and cell-stimulus interaction and allowed frequent supernatant sampling without cell culture disturbance. An in vitro microphysiological model of the human adipose tissue has been successfully demonstrated in the chip. The anatomical features of the model were demonstrated by mimicking the infiltration of the immune cells within the adipose tissue and the major functions of the cellular structure were also demonstrated, which provide a close physiological scenario that exhibits key characteristics of type 2 diabetic adipose tissues. Finally, in this paper, we demonstrated a microfluidic platform for long-term cell co-culture that is amenable for integration with various functional components for enabling precise control of the cell culture environment. For instance, a selected compartment (e.g., feeding compartment) can be filled with a sponge-like material [35] to allow the controlled release of nutrients/drugs to the neighboring cell culture. Local heat exchange can also be achieved by integrating a polymer-based heat exchanger that enables investigating the effect of local temperature on the cell–cell crosstalk [36]. In addition, in situ capture and detection of the secreted cytokines can be achieved by utilizing functionalized magnetic beads in a downstream fluidic compartment for time-resolved magnetic immune assay from the same set of cells, which will be the scope of our future studies. 

## 4. Conclusions

A planar compartmentalized microfluidic system with porous sidewalls (barriers) was designed, fabricated and tested. The microfluidic chip comprises several fluidic compartments which are physically separated by thin side walls. The side walls are characterized by a high-density array of small pores with a cross-sectional area smaller than the cell size. The compartmentalized structure enabled the co-culture of two types of cells with each cell type being cultured in its designated compartment. The fluidic compartments are arranged in the planar organization in close proximity to each other, which enables continuous perfusion/diffusion of chemical/biological signals between the cells in the different compartments in a dynamic manner that emulates the environment in the human tissue. The microfluidic system was tested by culturing human adipose tissue that is infiltrated with immune cells to study the role of the interplay between the two cells in type 2 diabetes. However, the system can be employed as a platform for hosting many organ/tissue models.

## Figures and Tables

**Figure 1 micromachines-11-00565-f001:**
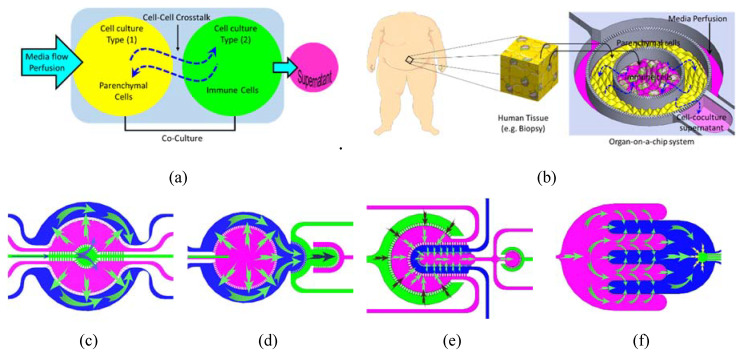
(**a**) Conceptual schematic drawing of the perfusion compartmentalized system with the proposed cell culture. (**b**) A biological model (co-culture) hosted within the perfusion system. (**c**–**f**) Fabricated and tested microfluidic chip showing suggested inter-compartment fluidic crosstalk.

**Figure 2 micromachines-11-00565-f002:**
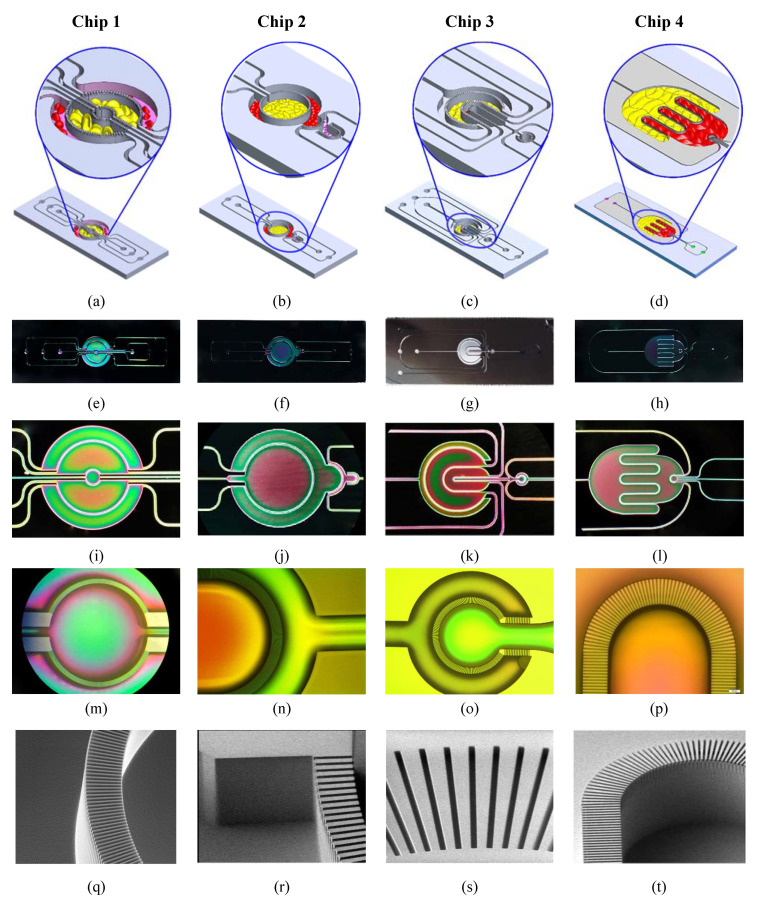
The fabricated and tested devices. (**a**–**d**) 3D schematic drawing of the chip with a detailed view of the compartmentalized structure. (**e**–**h**) Optical images of the fabricated devices. **(i**–**l**) Detailed views of the cell culture compartments. (**m-p)** Detailed views of key structures. (**q**–**t**) SEM images of the porous barrier.

**Figure 3 micromachines-11-00565-f003:**
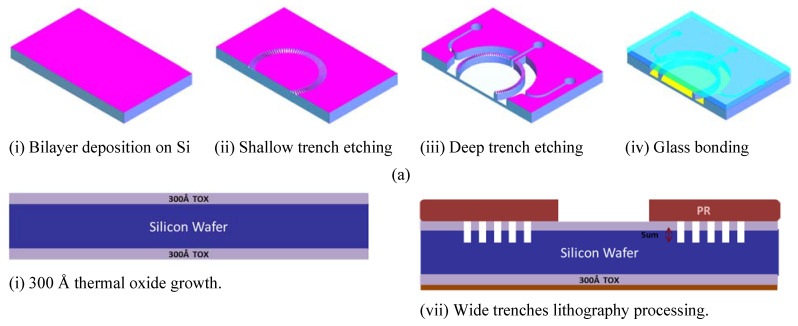
(**a**) The major fabrication processes of the microfluidic chips demonstrated in 3D drawing. (**b**) The process details in a cross-sectional view. (**c**) A schematic drawing of the fluidic set-up and perfusion circuit. The perfusion circuit inside the cell culture incubator while the driving pump is located outside the incubator. (**d**) An overview image of the perfusion circuit.

**Figure 4 micromachines-11-00565-f004:**
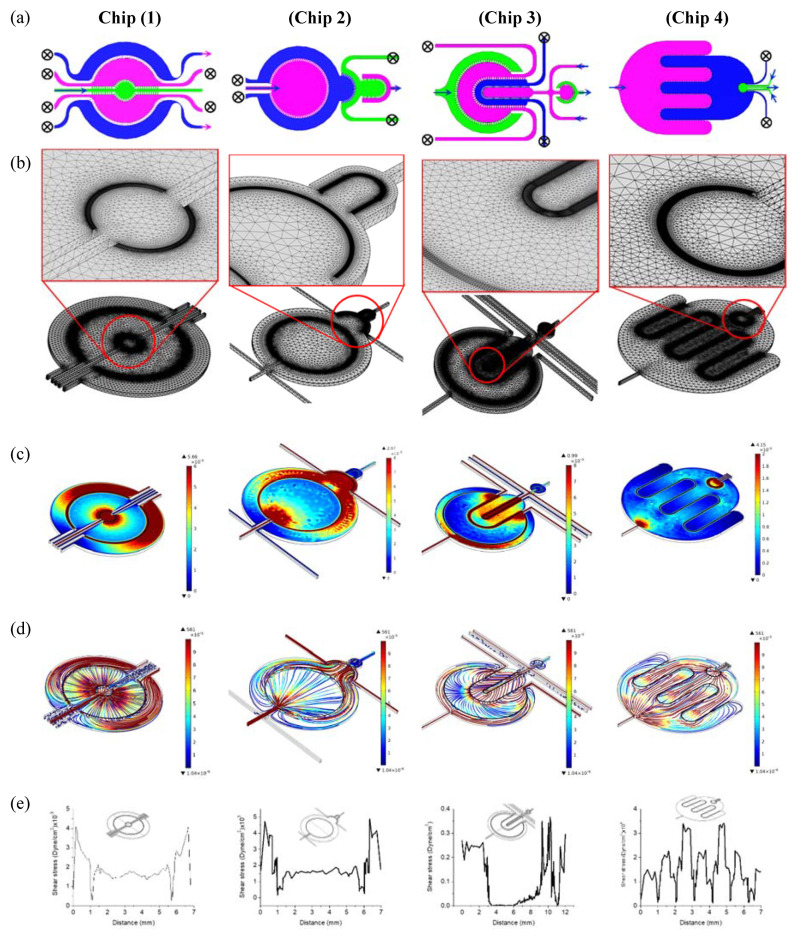
Finite element analysis of the flow profile within the four microfluidic chips at selected flow “in/out” configurations. (**a**) Flow conditions; (**b**) Meshed model with detailed zone of interest; (**c**) Velocity filed profile (mm/s); (**d**) Velocity profile (Streamlines); (**e**) Shear stress.

**Figure 5 micromachines-11-00565-f005:**
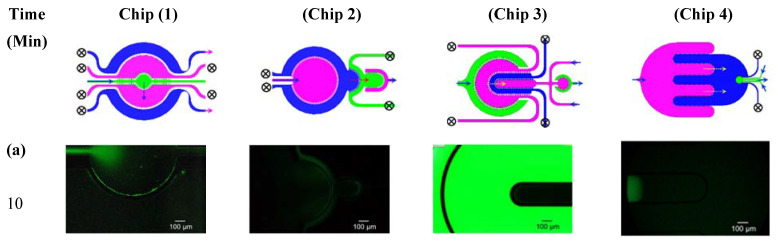
(**a**–**h**) Series of fluorescent images showing the inter-compartment transport of FITC-dextran at different time intervals over 90 min at a flow rate of 50 µL/min. (**i**) Fluorescence intensities due to the FITC-dextran permeation from one compartment to the adjacent one, as indicated by the dotted arrows.

**Figure 6 micromachines-11-00565-f006:**
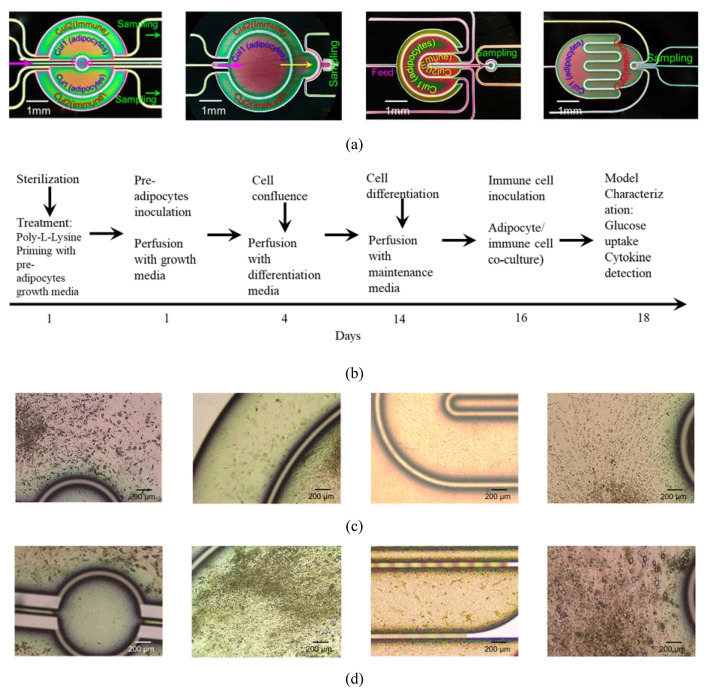
The sequence of adipocyte/immune cells co-culture and optical images of the co-culture over 24 days. (**a**) Chip images with the cell cultutrre designation within the corresponding compartments; (**b**) Summary of experemintal steps of the cell co-culture process; (**c**) Preadipocytes monoculture (5 days); (**d**) Differentiated adipocytes (14 days); (**e**) Adipocytes-immune cell co-culture (16 days); (**f**) Fully differentiated adipocytes (20 days); (**g**) Fully differentiated adipocytes (24 days).

**Figure 7 micromachines-11-00565-f007:**
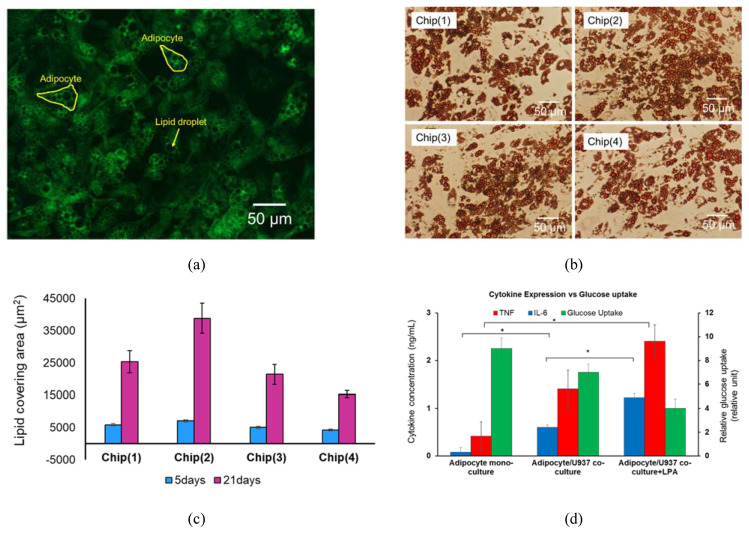
Adipocytes monocultured on-chip. (**a**) Fluorescence images of differentiated adipocytes after 14 days of differentiation medium treatment. (**b**) Lipid droplets within the adipoctes highlighted with Oil Red O staining. (**c**) Total lipid droplet coverage area in adipocyte culture compartments at 5 and 24 days after diffrentation medium treatment. (**d**) Glucose uptake vs. Il-6 and TNF-α expression at 24 h. Data were represented as mean ± SEM. * indicate statistical significance at *p* < 0.01.

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
