# Peer review of "OOCHIP: Compartmentalized Microfluidic Perfusion System with Porous Barriers for Enhanced Cell–Cell Crosstalk in Organ-on-a-Chip"

_micromachines, 2020, doi:10.3390/mi11060565_

Round 1

Reviewer 1 Report

The manuscript entitled "OOCHIP: Compartmentalized microfluidic perfusion system with porous barriers for enhanced cell-cell crosstalk in “organ-on-a-chip” describes a microfluidic-based organ-on-a-chip model. The work is very well designed, presented and explained. I am happy to recommend the work to be accepted following a minor revision, addressing my following comments:

Given you have used porous barriers to enable cell-cell communications, I would suggest you discuss the integration of highly porous sponges, as presented in Lab on a Chip, 2017 17 (14), 2517-2527, to your system to enable the release of chemicals:

Given the sensitivity of cell-cell crosstalk to temperature, please discuss the integration of convective heat exchangers, as presented in Analytical Chemistry, 2019, 91 (24), 15784-15790, to your system to enable studying cellular responses at desired static or transient temperatures.

Author Response

We thank the reviewer for his constructive comments and suggestions.  The following paragraph (Line 377-386) is added in the revised manuscript that addresses the reviewer suggestion:

Finally, in this paper, we demonstrated a microfluidic platform for long-term cell co-culture that is amenable for integration with various functional components for enabling precise control of the cell culture environment. For instance, a selected compartment (e.g. feeding compartment) can be filled with a sponge-like material [Thurgood et al, 2017] to allow the controlled release of nutrients/drugs to the neighboring cell culture. Local heat exchange also can be achieved by integrating a polymer-based heat exchanger that enables investigating the effect of local temperature on the cell-cell crosstalk [Zhue et al, 2019]. In addition, in situ capture and detection of the secreted cytokines can be achieved by utilizing functionalized magnetic beads in a downstream fluidic compartment for time-resolved magnetic immune assay from the same set of cells which will be the scope of our future studies. 

Reviewer 2 Report

This paper describes a set of compartmentalized microfluidic devices to permit co-culture while applying fluidic flow to mimic the environment in the human tissue.  The device consists of several fluidic compartments separated by porous sidewalls.  The authors demonstrate that this device can be used to culture human adipocytes with monocyte-like U937 cells.  Although this culture system may be useful, this study is very similar to their previous work (Ref. 9) in which they showed co-culture of adipocytes and U937 cells using a compartmentalized microfluidic device.  Although they show 4 types of devices, chip (1) is the exact same as their previous device reported in 2019 (Ref. 9) and all the devices are basically similar.  Since the device design and the cell types are same as what they showed in the previous paper, this paper doesn’t warrant publication with its current findings.  The authors have to clarify the difference between the previous work and the current study as well as new findings.

Author Response

We thank the reviewer for his/her comments.  However, we believe that our manuscript has been unfairly judged against our previous publication (Ref.9).  We are kindly asking the respected referee to have a look again at the manuscript carefully to find out that the scope of our manuscript is NOT to demonstrate that the device can be used to culture human adipocytes with monocyte-like U937 cells, but we are presenting a novel and carefully designed platform for organ-on-a-chip studies that permit efficient cellular interaction. The presented adipocyte-immune cell results are just to prove the usability of the system, hence, the biological data here were brief. Therefore, we must highlight that the scope of the current manuscript is totally different and focuses on the engineering (design, fabrication and flow profile) in four different OOC chips while, in our previous publication, the focus was to demonstrate the biological model with a very minimum discussion of the engineering aspect of the chip.

While we could exclude the design which was used in the previous publication we decided to include it for comparison and at the same time we avoided any overlapping of the presented results between the two manuscripts. We cited Ref.9 whenever is needed through the current manuscript.

In fact, in the current study, we presented a comprehensive analysis of the flow profile using both Finite Element Analysis and experimental methods.  We presented a detailed inter-compartment transport of FITC-dextran at different against time which clearly shows the possible way of inter-compartment cross-talk during cell culture hence shows how cells in the different compartments may communicate.

The data shown in our current manuscript are original, comprehensive, and substantial.  Our study has been carefully designed and implemented as judged by the other respected reviewers.

Reviewer 3 Report

OOCHIP: Compartmentalized microfluidic perfusion system with porous barriers for enhanced cell-cell crosstalk in "organ-on-a-chip."

Overall, it is a well-written article where authors provide an introduction and benefits of the organ-on-chip over the conventional cell culture models and animal models. To enhance the cell-cell interaction and enable co-culture of different cell types, they introduce a set of compartmentalized silicon-based microfluidic chips.  They use porous-barriers to mimic the in vivo-like paracrine exchange between cells in the human body. To further validate their model, they tested by culturing human adipose tissue and infiltrated them with immune cells to study their role in the contest of diabetes type 2. Moreover, the arrangements of the fluidic compartments in different compartments enabled continuous perfusion/diffusion of chemical/biological signals between the cells in a dynamic manner, which is more relevant physiologically.

However, I did find some issues which were not addressed. To complete this paper, I would advise the authors to address the following points.

  • They do not talk about other existing body-on-a-chip/ multi-organ-on-a-chip models. What are the advantages of this method compared to other existing co-culture models? Esch et al. 2016, Maschmeyer et al. 2015, Wagner et al. 2013, Kimura et al. 2015, and many more. It is better to explain their limitations to sell their work. 
  • One issue is with the figures labeling and numbering. It is confusing, with no appropriate directions or flow. (Especially in figure 2 (missed labeling) and 3). 
  • Figures do not have scale bars. In figure 6 (f, day 20 and g, day 21), there seem to be drastic changes in 1 day. Is it possible to observe such changes within one day? What are the possible reasons?
  • Figure 7 c and d. Are they statistically significant? 
  • "Adipocytes" have been misspelled as "Adispocytes" in several places. 
  • They used a porous barrier structure. How do authors ensure these are not obstructed during bonding or by unattached/dead cells floating around?

Author Response

We thank the reviewer for his/her constructive comments and suggestions.  Please see below our feedback:

They do not talk about other existing body-on-a-chip/ multi-organ-on-a-chip models. What are the advantages of this method compared to other existing co-culture models? Esch et al. 2016, Maschmeyer et al. 2015, Wagner et al. 2013, Kimura et al. 2015, and many more. It is better to explain their limitations to sell their work.

We added the following paragraph that addresses the state of the art of body-on-a-chip.  The text (Line 76 – Line 87) differentiates our system from the literature. 

“Modeling of human physiology and diseases with interconnected micro-physiological (MPS) or body-on-a-chip models, is emerging as a prominent frontier in medicine. The last few years witnessed several attempts to demonstrate complex MPSs with higher-order tissue functions [8-18]. Inter-organ communication is mainly studied through a systemic common medium that interconnects different organs modules to mimic the circulating blood which can mediate organ-organ crosstalk. Despite the tremendous achieved progress, the existing technology gap between the current MPS technology and in vivo conditions still wide. Particularly, the employed methods for emulating cell-cell cross talk are not capable to mimic those in vivo. The vast majority of OOC and MPS devices are fabricated using soft lithography techniques out of PDMS which absorb small organic compounds, including the precious analytes.  In addition, using these techniques, may not be able to achieve precise positioning of different types of cells in close proximity with a reliable and repeatable manner for efficient and physiologically relevant paracrine signaling.”

One issue is with the figures labeling and numbering. It is confusing, with no appropriate directions or flow. (Especially in figure 2 (missed labeling) and 3).

Figure 2 is arranged according to the 4 chips with the first row shows a 3D drawing of the chips (a-d) followed by detailed optical images of the 4 devices with different magnification, then SEM images of the same device.  Columns (1), (2), (3), (4) shows all drawing/images of Chip (1), (2), (3), (4) respectively.

Figure 3 shows all the information related to fabrication.  Despite its length, it arranged in three parts (a), (b), and (c) and easy to follow.

The display of data/images in Figures 4, 5, and 6 also follow the same style of Figure 2, with 4 columns and each column shows image/data related to one of the 4 chips.

Figures do not have scale bars. In figure 6 (f, day 20 and g, day 21), there seem to be drastic changes in 1 day. Is it possible to observe such changes within one day? What are the possible reasons?

All missing scale bares are added.

We thank the reviewer for his notification.  After carefully revising our images records we found that the images in (g) were taken on day 24.  This is now corrected in the caption. 

Figure 7 c and d. Are they statistically significant?

P-value is added.

"Adipocytes" have been misspelled as "Adispocytes" in several places.

Misspelled words are corrected.

They used a porous barrier structure. How do authors ensure these are not obstructed during bonding or by unattached/dead cells floating around?

We did not see the issue of obstruction of the porous barrier due to bonding.  However, all chips were carefully characterized prior to running the cell culture experiments to ensure that good inter-compartment flow through the porous barrier. To avoid obstruction during the cell seeding/culture, cell seeding was optimized at a sufficient cell number. After a few hours of cell seeding, the perfusion flow switched ON which help in preventing cells to obstruct the small pores.  Some cells do stuck but the perfusion flow still

Round 2

Reviewer 2 Report

This paper describes compartmentalized microdevices to emulate cell-cell interactions and functions of tissues.  However, this paper does not show any data to demonstrate how similar this system is to in vivo and whether or not this system is better than other systems.  The authors compared four types of their own devices, whereas they do not compare them with any other methods.  Sine the authors claim that current systems are different from in vivo conditions and have some limitations, they need to demonstrate how good this system is, compared with other systems.  Although the authors claim that this system can recapitulate physiological characteristics including 7 things, other organs-on-chips have the same advantages.  Importantly, although the authors claim that their device enables “efficient cellular interaction” and “mimicking the paracrine exchange between cells in the human body”, they do not demonstrate specifically how efficient and how similar to in vivo this system actually is.  There is no quantitative data to demonstrate this system is similar to in vivo conditions although this paper mainly claim that this system mimics the environment in the body.  By the way, I don’t see Figure 8 in this paper.  Therefore, this paper doesn’t warrant publication with its current findings.  The authors need to rewrite the entire manuscript if they want to claim a novel design platform of organs-on-chips.

Author Response

Dear reviewer, thank you for your comments.

This getting very confusing!

On 5 May 2020, I received the reviewer's reports (3 reviewers). The reviewers reports were studied carefully and addressed point by point. The manuscript has been revised taking into account all the reviewers comments and suggestions. The revised manuscript as well as our response to the reviewers comments have been uploaded into the system on 12 May 2020. We believe that our responses to the reviewers sufficiently addressed all the concern points raised by the three reviewers. We saved no effort to eliminate any limitations to bring the manuscript to an even more outstanding value.

Reviewer 3 Report

I am ok with the rebuttal and applied changes. it can be accepted now. 

Author Response

Thank you for your review.